# Genome Size of Life Forms of *Araceae*—A New Piece in the C-Value Puzzle

**DOI:** 10.3390/plants11030334

**Published:** 2022-01-27

**Authors:** Domen Kocjan, Jasna Dolenc Koce, Florian Etl, Marina Dermastia

**Affiliations:** 1Department of Biology, Biotechnical Faculty, University of Ljubljana, SI-1000 Ljubljana, Slovenia; kocjan_domen@hotmail.com; 2Department of Botany and Biodiversity Research, University of Vienna, A-1030 Wien, Austria; florian.etl@univie.ac.at; 3Department of Biotechnology and Systems Biology, National Institute of Biology, University of Ljubljana, SI-1000 Ljubljana, Slovenia; marina.dermastia@nib.si

**Keywords:** aquatic plant, aroid, epiphyte, image cytometry, nuclear DNA amount, terrestrial plant

## Abstract

The genome size of an organism is an important trait that has predictive values applicable to various scientific fields, including ecology. The main source of plant C-values is the Plant DNA C-values database of the Royal Botanic Gardens Kew, which currently contains 12,273 estimates. However, it covers only 2.9% of known angiosperm species and has gaps in the life form and geographic distribution of plants. Only 4.5% of C-value estimates come from researchers in Central and South America. This study provides 41 new C-values for the aroid family (*Araceae*), collected in the Piedras Blancas National Park area in southern Costa Rica, including terrestrial, epiphytic and aquatic life forms. Data from our study are combined with C-value entries in the RBGK database for *Araceae.* The analysis reveals a wider range of C-values for terrestrial aroids, consistent with other terrestrial plants, a trend toward slightly lower C-values for epiphytic forms, which is more consistent for obligate epiphytes, and comparatively low C-values for aquatic aroids.

## 1. Introduction

Genome size is a term widely used in the scientific literature, but it is used with different meanings depending on the author [1]. We use it here as a term that expresses the amount of DNA in the nucleus of a cell. The nuclear DNA amount is usually expressed as what is called the C-value, where “C” stands for “constant” [2]. The 1C-value refers to the size of the haploid genome, regardless of chromosome numbers and degree of polyploidy. Such a genome is found in an unduplicated nucleus of a haploid gamete. The 2C-value refers to the nuclear DNA amount in a duplicated nucleus of a gamete and an unduplicated nucleus of a diploid somatic cell. The 4C-value refers to the amount of nuclear DNA in a duplicated diploid nucleus before mitosis or meiosis. C-values higher than 4C are results of endopolyploidy [3], with the highest known multiplication being 24,576 C, as measured in the endosperm of the aroid *Arum maculatum* [4].

The C-value varies greatly among species. There is no correlation between the complexity of an organism and its genome size. This was known as the “C-value paradox” before the discovery of noncoding DNA [5,6]. Although knowledge of the importance of genome size has improved in recent decades, the source of the extensive variation in nuclear genome size among eukaryotic species and the functional significance of the genome size remain largely unsolved. This puzzle is known as the “C-value enigma”, as described by Gregory [7].

Currently, there are three main hypotheses that attempt to explain the importance of genome size variations in organisms. The first hypothesis is that the variation is a random effect of mutations that duplicate and accumulate so-called nonessential or “selfish” DNA [8]. Species with larger genomes are thought to be those that better tolerate the accumulation of “selfish” DNA. Therefore, genome size, and the resulting larger cell size, are not of biological importance [6]. The second hypothesis is the so-called nucleotype hypothesis. The term nucleotype was formed as a counterpart to genotype. It is designed to describe the influence of the nucleus on the phenotype, which is based on the size and structure of the DNA, independent of the genes encoded in the DNA [9]. According to this hypothesis, genome size has an active biological role in an organism, and its life cycle is particularly correlated with genome size. The final hypothesis, called the nucleoskeletal hypothesis, states that cell size has an active biological role while genome size is a passive consequence of cell growth. A larger amount of DNA would primarily have the role of a nucleoskeleton to help to maintain an optimized ratio between the nucleus and cytoplasm, thereby optimizing cell processes [10,11].

Genome size is useful in a variety of disciplines, from molecular biology, taxonomy, evolutionary biology, phylogeny and ecology, to conservation [12]. More than 20 years ago, it was proposed that the nuclear DNA amount in vascular plants can be used to predict vegetation responses to landscape and climate change by linking genome size variations to ecological patterns and processes [13]. To test predictions of vegetation responses to climate and land use at different geographic scales, a need for appropriate data collection was also identified at that time. Based on the idea of improving the accessibility of plant genome sizes by making them available for reference purposes in the form of published reference lists and/or on the Internet in a database of plant DNA C-values, an open-access database that is maintained by the Royal Botanic Gardens Kew, UK (RBGK) [14], was released in 2001.

The RBGK database currently contains 12,273 C-value estimates from different plant groups [14] and are expressed as the number of base pairs (bp) or as the mass of DNA in picograms (1 pg = 10^−12^ g) where one picogram is approximately 10^9^ base pairs [15]. In angiosperms, the C-values range from 1C = 0.06 pg to 1C = 150 pg. However, current C-value estimates represent only 2.9% of the C-values of the 369,434 known angiosperm species [16]. In addition to taxonomic inadequacies, this knowledge of genome size includes geographic inadequacies. In 2011, Bennett and Leitch [17] showed that 58.2% of all genome data were reported by European researchers, with only 4.5% from Central and South America. There is also insufficient data for certain life forms, such as epiphytes.

Aroids (Araceae) are the third largest family of monocots, which comprise ca. 140 genera [18] with around 4000 species described [19]. These are spread all over the globe (Figure 1), but over 90% of species are found in the tropics [20]. The most diverse regions are Central and South America, Southeast Asia, the Malay Archipelago and continental tropical Africa. Most species are found in the tropics of Central and South America. Many species are endemic, while some are widespread [21]. Members of the aroid family are found in a variety of different habitats, from tropical dry to pluvial rainforest, through tropical swamps and cloud forests, to subarctic marshes and montane plains [22].

Most aroids are terrestrial. Many are geophytes with spreading rhizomes [21,22] or climbers that require external support [23]. However, both geophytes and climbers require a root connection to the soil.

A common life form of aroids is epiphytes. These plants grow on a host plant and do not require ground contact to survive. Epiphytes obtain all of their required nutrients and water from precipitation and leaf litter. The so-called litter-basket epiphytes can collect water and leaf litter due to their shape and leaf arrangement [21]. Among the aroids, there are obligate epiphytes (e.g., *Anthurium obtusum*), which are exclusively epiphytic, and facultative epiphytes (e.g., *Philodendron auriculatum*), which can also grow on the ground [24]. The facultative epiphytes thrive in habitats such as cloud forests, where living conditions on the ground and on host plants are similar. Hemiepiphytic aroids (e.g., *Anthurium gymnopus*) germinate as epiphytes, then sprout feeder roots that reach the ground [21,25]. Some studies have also recognized secondary hemiepiphytes, although more recent studies [23,24] recommend the term nomadic vines or nomadic climbers (a term that is further used in this review). These germinate as terrestrial plants, then begin climbing on a host plant and eventually lose contact with the soil as their initial roots die (e.g., *Monstera adansonii*). The plant only remains connected to the soil through its adventitious roots [23,24,26].

Depending on how much they depend on constant contact with water, various life forms of aroids can be considered aquatic. They can be floating aquatic plants (e.g., *Lemna minor*) or submerged aquatic plants (e.g., *Cryptocoryne crispatula*). These grow in water bodies and can be free-floating or rooted in the substrate, whereas rheophytes (e.g., *Anthurium amnicola*) grow on rocks in waterways, and helophytes (e.g., *Orontium aquaticum*) are wetland plants that grow in waterlogged soil [21]. This becomes more complicated when it is considered that epiphytic plants can also grow on rocks rather than host plants. These are epilithic, but this form is optional in virtually all juvenile epiphytes. Rheophytes can also be considered epilithic, although they are also aquatic [22]. 

Of the approximately 4000 known aroid species [19], C-values are known for only 3.7% of them, and most of them are already included in the RBGK C-value database [14]. While the number of C-value estimates in the RBGK C-value database (Figure 1) covers most aroids in Africa, Asia and Australia, it is overestimated in Europe and North America, likely due to repeated measurements of the C-value in the same plant. In comparison, the number of C-value entries in the database is 10 times lower (i.e., 35%) for Central and South America than would be expected given the number of aroids in the area.

To fill an important part of the identified life-form-specific and geographic gaps in the C-value database for aroids, the objective of this work was to estimate the C-value of terrestrial and various epiphytic forms of aroid species and genera not currently included in the database and originating from the area of Piedras Blancas National Park in southern Costa Rica, which is particularly rich in aroids [27]. The new data were analyzed together with the existing data in the C-value database to reveal a possible relationship between the C-value and a specific life form of the aroids.

## 2. Results and Discussion

### 2.1. C-Value Estimates

Forty-one new C-values from six aroid life forms (Figure 2) were estimated (Table 1). These include 26 epiphytic aroid forms, which represent 57.8% of all of the data on epiphytic aroids previously included in the RBGK C-value database [14]. New C-value estimates include those of five genera measured for the first time (*Adelonema*, *Aglaonema*, *Dracontium*, *Rhodospatha*, *Stenospermation*), as well as data from wild forms of the genus *Spathiphyllum*, for which C-values were previously measured only for their horticultural cultivars [28,29]. We also measured C-values already recorded in the database [14] for obligatory epiphytes *Anthurium clavigerum* and *A. obtusum*, a facultative epiphyte *A. hoffmanii*, three terrestrial species *Alocasia longiloba,*
*Anthurium ochranthum* and *Xanthosoma sagittifolium*, a nomadic vine *Syngonium podophyllum* and an aquatic species *Pistia stratiotes*.

The new data were combined with the C-values from the RBGK database (Appendix A) and for the terrestrial *Amorphophallus konjac* with the value measured by Zhao et al. [30]. As there are some discrepancies between the genus and species names of plants in the RBGK database and the current taxonomy [18,31,32], we matched 18 synonyms with their current valid taxonomic status (Appendix A). For most species, only a newer synonym was assigned, but some species were transferred to other genera. Accordingly, the genus *Landoltia* has been excluded from the list here, and the genus *Thaumatophyllum* has been added. In the RBGK database, there were also some species with duplicate measurements because some species had C-value data stored under both their old and new names [14]. If the same author measured the C-value of a species twice under different names, one of them was excluded from the analysis. However, if two different authors measured the same species under different synonyms, we considered both C-values.

### 2.2. Aroid C-Values Related to Plant Life Forms

Analysis of the data from this study combined with the C-values from the RBGK database (Appendix A) showed a moderately wide range of 2C-values among the aroids, which averaged out as 2.95 pg DNA for the aquatic aroids, and 15.26 pg DNA for the terrestrial aroids, with a trend toward an interestingly different distribution of 2C-values among the different aroid life forms (Figure 3).

Bennett [33] has shown that perennial herbs have a wide range of C-values, which is also mirrored for the terrestrial aroids in the present study. Their mean value of 15.26 ± 10.28 pg DNA (Figure 3) is greater than 76% of all of the C-values analyzed, and ranks in the fourth quartile of aroid genome size. However, Grubbs’ and ROUT tests revealed two data points that were significantly different from other observations for the terrestrial life forms, namely, the 2C-values of *Aglaonema* cf. *marantifolium* (2C = 62.48 pg) (Table 1) and *Zamioculcas zamiifolia* (2C = 48.10 pg) [29].

Plants of different epiphytic types had similar 2C-values, which ranged from 8.23 ± 5.63 pg to 10.73 ± 5.56 pg (Table 1, Figure 3). In the group of facultative epiphytes, the analysis with Grubbs’ and ROUT tests revealed only one outlier, the 2C-value of *Anthurium grande* (2C = 27.05 pg) [34]. In the nomadic vine group, *Scindapsus pictus* (2C = 23.51 pg) was an outlier (Table 1) [34]. Obligatory epiphytes showed a more uniform distribution of 2C-values (i.e., 9.20 ± 2.89 pg), with no major outliers. Whether this uniformity of C-values in epiphytic aroids is of any biological significance is not clear at present.

The aquatic forms had the smallest C-values among the aroids (i.e., 2.95 ± 5.04 pg), and their C-values represented a small range at the lower end of the scale (Table 1, Figure 3). The genus *Wolffia* has 9 to 11 floating species that include the smallest flowering plants on Earth, and these had the highest C-values among the aquatic aroids (mean 2C = 2.27 pg) [35]. The C-values of species in the rooted genus *Cryptocoryne* were only slightly lower (mean 2C = 1.83 pg) [29]. The Grubbs’ and ROUT tests detected some deviations from the mean value for the aquatic aroids. Two of these values were 2C = 10.36 pg and 2C = 9.83 pg, for *Anthurium amnicola* and *Anthurium antioquiense*, respectively (Appendix A). Despite their rheophytic lifestyle in nature, they are also cultivated as terrestrial ornamentals, and their C-values were within the range of the large diversity of life forms within the entire genus *Anthurium*. Another outlier among the aquatic aroids was *Orontium aquaticum*, with a 2C value of 30.00 pg DNA [14]. The low C-values of aquatic aroids might be related to the prevalence of rapid vegetative reproduction [36,37], which can promote opportunistic growth during episodes of suitable growing conditions [13].

The genome size of the aroids was also analyzed from a phylogenetic perspective (Figure 4). The analysis showed that C-values are generally evenly distributed among life forms and are not related to the phylogenetic position. The low C-values of genera with the aquatic life form were confirmed. However, the high C-value of *Orontium aquaticum* might be related to its distinct morphological and phylogenetic lineage (Figure 4).

## 3. Conclusions

The very different living conditions under which most aroids grow allow for very diverse life forms (this study, [21,22,23,24,26]). As there are no sharp boundaries between some of the aroid life forms, e.g., terrestrial and epiphytic, the similar range of C-values among them is not unexpected and was confirmed in this study with the addition of 41 new C-value estimates from Costa Rica to the analysis, which included epiphytes. However, further studies are needed to determine whether the small differences in C-values have specific ecological significance. On the other hand, the low C-values of aquatic forms might indicate their ecological role, possibly related to one of the C-value puzzle hypotheses [39].

## 4. Material and Methods

### 4.1. Plant Material

The samples of *Araceae* were collected in the vicinity of La Gamba tropical station, Costa Rica (8°42′03.4″ N, 83°12′05.9″ W), mainly in primary and secondary lowland moist forest, along trails and on the banks of forest streams. Some samples were collected along forest edges and in roadside clearings. A total of 100 individuals of 41 species were collected and were identified according to Croat [40,41,42], Grayum [43,44], Cedeño-Fonseca et al. [45] and Weissenhofer and Grayum [46]. The herbarium vouchers of collected specimens are deposited in the Herbarium WU (Department of Botany and Biodiversity Research, University of Vienna, Vienna, Austria).

We used different tissue of collected aroids for genome size analysis, primarily meristemic tissue, such as root tip, shoot tip, axillary bud, inflorescence or ovule. Young leaves or other parts were used when meristematic tissue was not available (Table 1). The tissue was dissected from the plant, fixed in 4% formaldehyde in Sørensen buffer (pH 6.8) for 90 min at ambient temperature, post-fixed in several changes of 3:1 methanol:acetic acid for at least 24 h at 4 °C and finally transferred in 96% ethanol for long-term storage at −20 °C and transport [47].

### 4.2. Genome Size Measurement

The genome size data were obtained by DNA image cytometry of fixed and Feulgen stained tissues, as described previously [47,48]. Briefly, fixed tissue was hydrolyzed in 5 M HCl for 90 min at 20 °C, then stained with Feulgen reagent for 120 min at 20 °C or overnight at 4 °C and washed in several changes of SO_2_-water. Squash preparations were conducted in 45% acetic acid and frozen on the microscope slide. The root tips of calibration standard *Pisum sativum* cv. ‘Kleine Rheinländerin’ were stained simultaneously with aroid samples.

The amount of nuclear DNA was measured densitometrically using interphase-peak DNA image cytometry [48]. The integrated optical density was measured for approx. 150–250 interphase nuclei per slide. The 2C-values of aroid specimens were calculated as estimates in pg DNA using *Pisum sativum* 2C = 8.84 pg as the standard calibration value [49].

### 4.3. Statistical Analysis

The data were statistically analyzed to calculate mean and median values, standard errors and the coefficient of variation, using Excel (Microsoft) and GraphPad Prism v8.3.4 software. Grubb’s and ROUT tests were used for outliers to test if any of the species was an outlier within its life form.

The phylogenetic tree was acquired from http://www.timetree.org (accessed on 7 January 2022) [50] and based on data from Nauheimer et al. [51], with additional data for genera *Adelonema* and *Thaumatophyllum* added according to Vasconcelos et al. [52]. The tree was modified with software MEGA X v10.2.4 [53].

## Figures and Tables

**Figure 1 plants-11-00334-f001:**
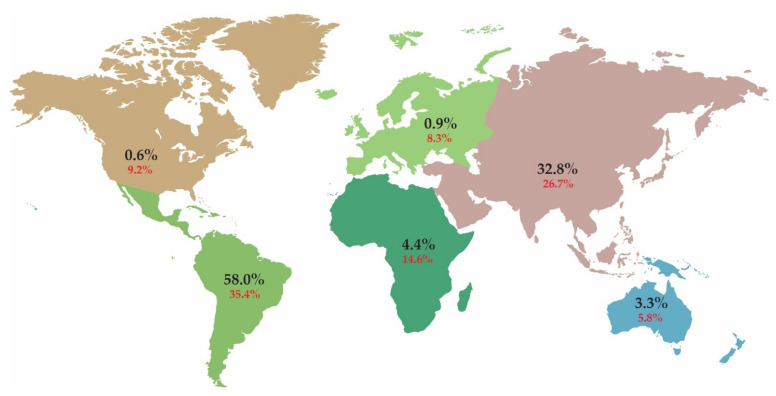
The global distribution of *Araceae*. Values in black indicate the percentage of aroid species on each continent, with distribution data taken from Plants of the World Online [18]. Values in red indicate the percentage of known aroid C-values for each continent, with data from the RBGK database [14].

**Figure 2 plants-11-00334-f002:**
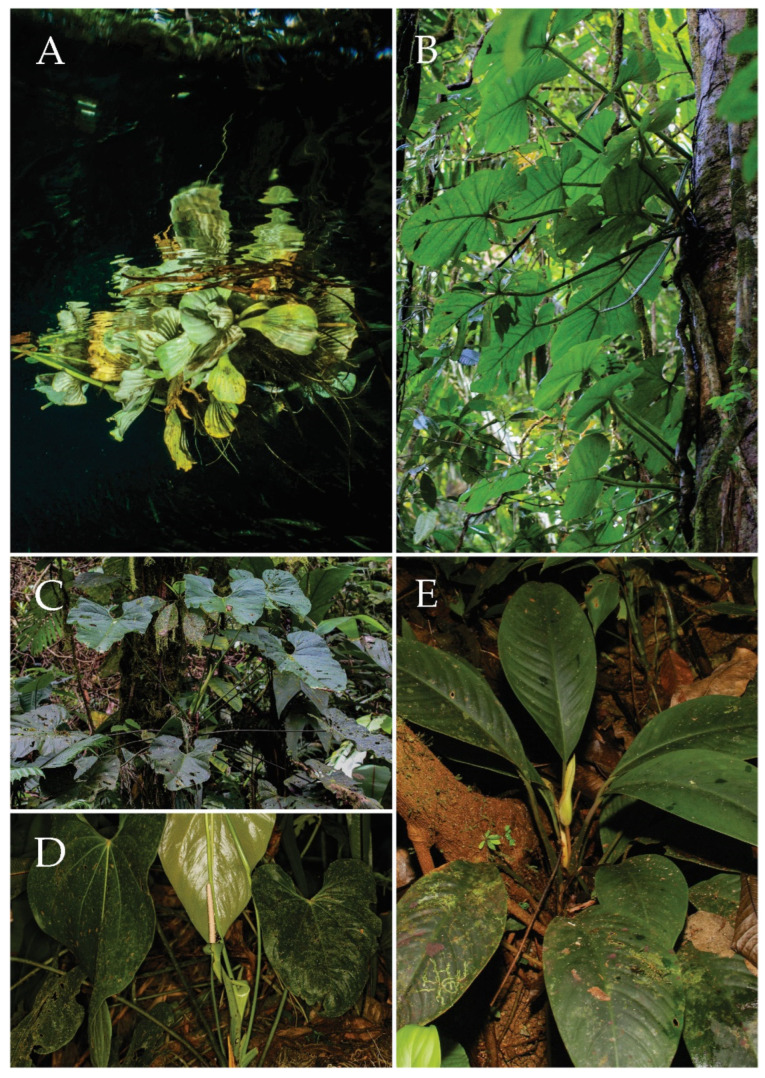
Life forms of the *Araceae* collected in the Piedras Blancas National Park area: (**A**) Aquatic plant (*Pistia stratiotes*); (**B**) Nomadic vine (*Philodendron pterotum*); (**C**) Obligatory epiphyte (*Anthurium ravenii)*; (**D**) Facultative epiphyte (*Anthurium hoffmanii*); (**E**) Terrestrial plant (*Adelonema erythropus*).

**Figure 3 plants-11-00334-f003:**
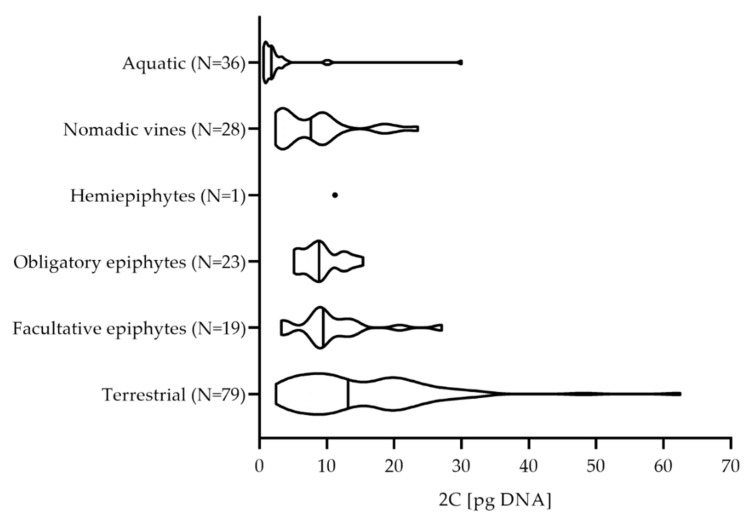
Violin plot showing the distribution of the 2C-values for each aroid life form. Each violin ranges from the minimum to the maximum value, and its shape represents the frequency distribution of the data. The lines mark the median values calculated for the number of measured samples (N).

**Figure 4 plants-11-00334-f004:**
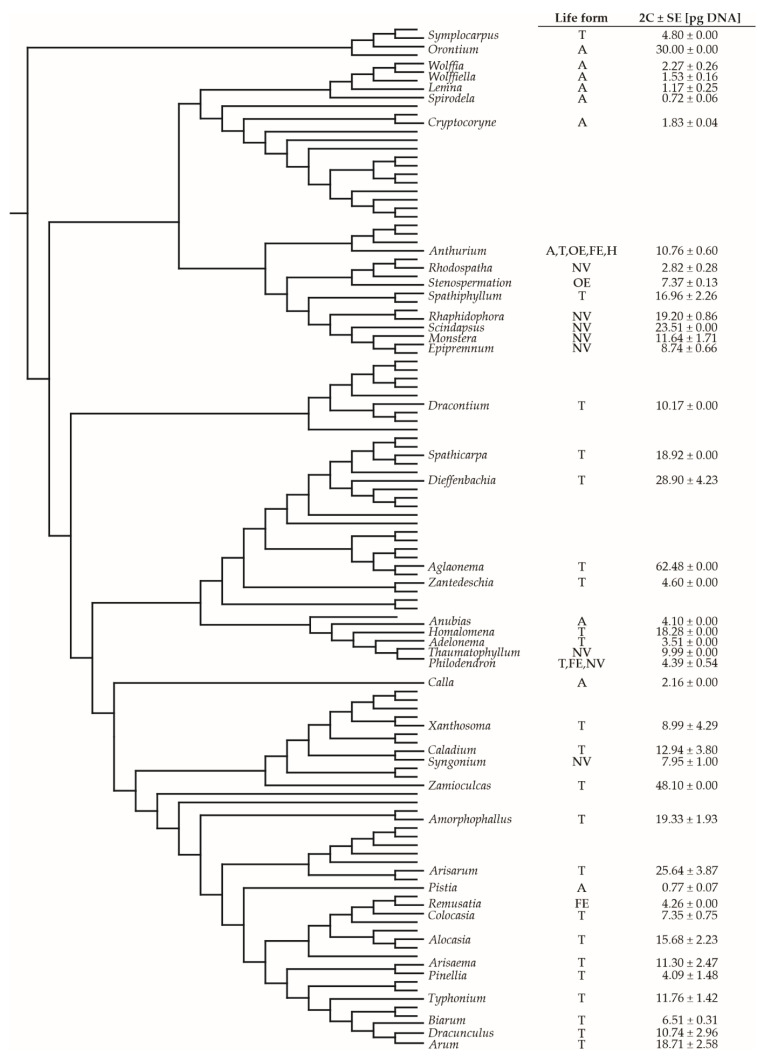
The phylogenetic tree of Araceae according to the Angiosperm Phylogeny Group system [38] with life forms and genome size data from our study and RBGK database [14]. Life forms are indicated with letters: A (aquatic), T (terrestrial), OE (obligatory epiphyte), FE (facultative epiphyte), H (hemiepiphyte), NV (nomadic vine). 2C-values are presented as means ± standard error (SE).

**Table 1 plants-11-00334-t001:** Details of the aroids collected in the Piedras Blancas National Park area, with information on their life form and genome size measurements. The mean 2C-value, standard error (SE) and coefficient of variation (CV) were calculated for the numbers of measured samples (N).

Genus	Species	Life Form	Tissue Used for C-Value Measurement	N	2C ± SE [pg DNA]	CV [%]
*Adelonema*	*A. erythropus*	Terrestrial	Leaf, unknown	9	3.50 ± 0.12	10.7
	*A. wendlandii* ^a^	Terrestrial	Meristem tissue, unknown, inflorescence, stem (cap)	12	3.51 ± 0.08	7.8
*Aglaonema*	*A.* cf. *marantifolium*	Terrestrial	Unknown	2	62.48 ± 2.70	6.1
*Alocasia*	*A. longiloba* ^b^	Terrestrial	Leaf, stem	3	27.50 ± 0.70	4.4
	*A. clypeolata*	Terrestrial	Young leaves, inflorescence, inflorescence (seed embryo)	7	11.02 ± 0.30	6.7
*Anthurium*	*A. brownii*	Obligatory epiphyte	Stem	8	8.45 ± 0.10	4.3
	*A. clavigerum* ^b^	Obligatory epiphyte	Unknown, stem	16	13.23 ± 0.15	4.7
	*A. hacumense*	Obligatory epiphyte	Roots, inflorescence	11	8.09 ± 0.20	8.2
	*A. hoffmanii* ^b^	Facultative epiphyte	Unknown	9	7.83 ± 0.20	7.1
	*A. obtusum* ^b^	Obligatory epiphyte	Unknown, inflorescence, inflorescence (seed embryo)	9	5.54 ± 0.10	7.7
	*A. ochranthum* ^b^	Terrestrial	Roots, leaf, stem, stem (cap)	6	12.48 ± 0.30	5.3
	*A.* cf. *ravenii*	Obligatory epiphyte	Unknown	4	8.63 ± 0.20	3.5
*Dieffenbachia*	*D. concinna*	Terrestrial	Stem (bud)	3	33.13 ± 0.30	1.7
*Dracontium*	*D. pittieri*	Terrestrial	Unknown	2	10.17 ± 0.10	1.7
*Monstera*	*M. adansonii*	Nomadic vine	Stem	3	10.54 ± 0.46	9.1
	*M. gambensis*	Nomadic vine	Stem (bud)	3	9.21 ± 0.20	3.6
	*M. pinnatipartita*	Nomadic vine	Leaf, unknown, stem (cap)	8	12.04 ± 0.30	6.6
*Philodendron*	*P*. *auriculatum*	Facultative epiphyte	Unknown, inflorescence, stem	10	3.23 ± 0.10	8.4
	*P*. *fragrantissimum*	Nomadic vine	Leaf, stem	10	4.86 ± 0.10	7.7
	*P*. *grandipes*	Terrestrial	Inflorescence, inflorescence (seed embryo), stem	6	4.30 ± 0.10	4.8
	*P*. *mexicanum*	Nomadic vine	Leaf, unknown	9	3.65 ± 0.20	15.1
	*P*. *microstictum*	Nomadic vine	Inflorescence	8	3.56 ± 0.00	3.4
	*P*. *opacum*	Nomadic vine	Leaf, inflorescence (seed embryo)	7	3.07 ± 0.10	4.3
	*P*. *platypetiolatum*	Nomadic vine	Leaf, stem	9	3.55 ± 0.10	6.4
	*P*. *popenoei*	Terrestrial	Unknown, stem	6	2.90 ± 0.00	2.7
	*P*. *pterotum*	Nomadic vine	Unknown, stem (cap, bud)	7	5.12 ± 0.10	6.7
	*P*. *sagittifolium*	Nomadic vine	Leaf (leaf embryo), stem, stem (cap)	9	3.60 ± 0.00	3.5
	*P*. *rhodoaxis*	Nomadic vine	Stem, root	6	2.54 ± 0.10	12.8
	*P*. sp.	Facultative epiphyte	Leaf, root	3	3.28 ± 0.10	7.8
	*P. tripartitum*	Nomadic vine	Leaf, stem	7	3.78 ± 0.10	5.5
*Pistia*	*P. stratiotes* ^b^	Aquatic	Leaf, stem, stem (cap)	15	0.83 ± 0.00	8.7
*Rhodospatha*	*R. osaensis*	Nomadic vine	Root, leaf, stem	11	3.09 ± 0.27	28.9
	*R.* cf. *osaensis*	Nomadic vine	Leaf (leaf embryo)	4	2.54 ± 0.00	3.3
*Spathiphyllum*	*S*. cf. *leave*	Terrestrial	Leaf	5	22.12 ± 0.30	2.6
	*S. silvicola*	Terrestrial	Leaf, inflorescence	6	18.29 ± 0.50	6.7
	*S. wendlandii*	Terrestrial	Leaf, stem (node)	6	20.07 ± 0.40	5.1
*Stenospermation*	*S. angustifolium*	Obligatory epiphyte	Unknown, inflorescence, inflorescence (seed embryo), stem	9	7.49 ± 0.20	6.5
	*S*. cf. *maranthifolium*	Obligatory epiphyte	Inflorescence, stem	9	7.24 ± 0.20	9.1
*Syngonium*	*S. hastiferum*	Nomadic vine	Roots, leaf, unknown, stem	19	7.16 ± 0.20	9.2
	*S. podophyllum* ^b^	Nomadic vine	Roots, leaf, stem (cap)	10	5.48 ± 0.10	5.5
*Xanthosoma*	*X. sagittifolium* ^b^	Terrestrial	Leaf, stem	9	4.81 ± 0.10	6.1

^a^ The first chromosome number report for *Adelonema wendlandii* (2n = 10). ^b^ Primary estimate already in the Royal Botanic Gardens Kew database.

## Data Availability

The source data for this study are provided in Plant DNA C-values Database; Release 7.1, April 2019. Leitch, I.J., Johnston, E., Pellicer, J., Hidalgo, O., Bennett, M.D. https://cvalues.science.kew.org/ (accessed on 7 January 2022), and the references in this article.

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
