# Peer review of "Genome Size of Life Forms of *Araceae*—A New Piece in the C-Value Puzzle"

_plants, 2022, doi:10.3390/plants11030334_

Round 1
Reviewer 1 Report
Review plants-1553320 Kocjan et al. Araceae
In their review (?) the authors analyse the C-value data present in the C-value database of the Royal Botanical Gardens Kew for Araceae. Further genome sizes of 41 accessions were added to fill in gaps in the dataset. Allover, a wider range of C-values was detected for terrestrial aroids while epiphytic aroids had a tendency to have smaller genomes. This was also true for aquatic aroids. This manuscript was submitted as a review. However, to me this manuscript is neither a clear review nor a research paper.
The first section offers a good background on genome size estimation and offers also a bit of information on the database on which this review is based. Allover the text is written well and easy to follow. But considering it is supposed to be a review I think a lot of information is lacking.
Just as a small comment on a statement in the first section:
line 35-36: „it is more convenient to express this in picograms“
I do not really agree with this statement. This strongly depends on what data you are analysisng. If I sequence a genome I would never state the genome size in pg but always state it in Mbp. On the other hand, flow cytometry cannot give me a number of base pairs, only a rough value by conversion of the fluorescence to pg and then to bp. This is fine, but I don‘t think you can have a statement such as the one above in an introduction.
As there were new data added and statistical analyses performed I expected some kind of materials and methods section, may it be in the text or as a supplementary file, if it is not allowed to have materials and methods in a review.
However, the only time the method used for genome size estimation is used is in the header of table 1. There is also no description of the statistical method, it only appears in the legend of figure 1. There is no description of the statistical tests done (line 200,…). There is no mentioning of total number of samples analysed etc. If I want to get this information I need to sum up the numbers in fig. 1. Even if this is a review of some kind, I cannot judge from the text if the new data actually integrate well with the old data, are methods comparable etc.
Section 4 tries to relate C-values to life form by grouping the analysed data into life form categories and calculating their average C-values. Each passage stating the average C-values and some outliers in each group is finished by a small sentence on the possible meaning of the larger or smaller genome size. This reads more like an integrated results and minimal discussion section – but not like a review. I would have expected comparisons to other plant groups etc. which are mostly lacking.
The conclusion is also very small and ends very vague.
I don‘t see how this can be a review paper, I would have expected much more information – but it is also not a research paper as there is no hypothesis or clear aim that this analysis is following as well as no description of applied methods.
Reviewer 2 Report
The review entitled “Genome size of Life Forms of Araceae - a New piece in the C-value puzzle ” by Kocjan et al summarizes an analysis of the C-value estimates in the database for the aroid family– Araceae.
C-value refers to the amount of DNA in an unduplicated haploid cell, regardless of chromosome number and degree of polyploidy. Although the current plant C-value database of the Royal Botanic Gardens Kew contains 12,273 estimates, it covers only 2.9% of known angiosperm species and only 4.5% for those from Central and South America.
To fill in a gap in entries of specialized life forms– epiphytes, the authors add 41 new C-values to the analysis for the aroid family (Araceae) from southern Costa Rica, which reveal a wider range of C-values for terrestrial aroids. These new data fit a trend toward slightly lower C-values for epiphytic forms and comparatively low C-values for aquatic aroids.
Overall, the review presents new data analysis and fill in a gap in the plant database. The method cited is thorough. Conclusions are appropriate, and supported by the data. Data is available either within the manuscript. The review is sound, and I recommend accepting it in the present form.
Author Response
There are no comments of rev.2.
Reviewer 3 Report
This is review manuscript summarized and studied the genome size of many aroid species. Although this manuscript is listed as a review, the authors added some new data. Generally speaking, I am satisfied with this version. There are only several minor points that should be considered.
- In section 2, the authors summarized the distribution of Aroids all of the world. I feel it will be clearer if a global map with Aroids distribution percentage in different continents can be added in this part.
- Also, in section 2, the authors described the main identifier of the aroid family and different life forms of the aroid family. I hope the authors can provide some typical pictures of aroid flowers, common life forms with their growth environments.
- In section 3, please add several sentences to describe how the C values are measured.
- The references format is not consistent, please check them carefully.
Round 2
Reviewer 1 Report
All previous comments have been adressed adequately. The manuscript has improved significantly as a research manuscript. Materials and methods section is now present.
Still I think the contribution of the paper to current knowledge is rather small as it only includes a download of data, generation of 41 genome size estimates and calculation of average values for life forms. This looks more like a small student project rather than a full reesearch manuscript.
I think it is important to add new genome size estimates and they should be published. But I doubt there are sufficient data and analyses inhere for a full research paper.
Author Response
The manuscript was improved by adding the phylogenetic perspective of genome size in Araceae. A new Fig. 4 is added. The analysis showed that C-values are generally evenly distributed among life forms and are not related to the phylogenetic position. The low C- values of genera with the aquatic life form were confirmed. However, the high C-value of Orontium aquaticum might be related to its distinct morphological and phylogenetic lineage.